# Human and Pig Pluripotent Stem Cells: From Cellular Products to Organogenesis and Beyond

**DOI:** 10.3390/cells12162075

**Published:** 2023-08-16

**Authors:** Yiyi Xuan, Björn Petersen, Pentao Liu

**Affiliations:** 1Stem Cell & Regenerative Medicine Consortium, School of Biomedical Sciences, Li Ka Shing Faculty of Medicine, The University of Hong Kong, Hong Kong, China; xuanyiyi@connect.hku.hk; 2Institute of Farm Animal Genetics, Friedrich-Loeffler-Institut, Mariensee, 31535 Neustadt am Rübenberge, Germany; bjoern.petersen@fli.de; 3Center for Translational Stem Cell Biology, Hong Kong, China

**Keywords:** pluripotent stem cells, regenerative medicine, cellular products, xenotransplantation, interspecies chimera

## Abstract

Pluripotent stem cells (PSCs) are important for studying development and hold great promise in regenerative medicine due to their ability to differentiate into various cell types. In this review, we comprehensively discuss the potential applications of both human and pig PSCs and provide an overview of the current progress and challenges in this field. In addition to exploring the therapeutic uses of PSC-derived cellular products, we also shed light on their significance in the study of interspecies chimeras, which has led to the creation of transplantable human or humanized pig organs. Moreover, we emphasize the importance of pig PSCs as an ideal cell source for genetic engineering, facilitating the development of genetically modified pigs for pig-to-human xenotransplantation. Despite the achievements that have been made, further investigations and refinement of PSC technologies are necessary to unlock their full potential in regenerative medicine and effectively address critical healthcare challenges.

## 1. Introduction

Pluripotent stem cells (PSCs) possess extensive development potential and propagation abilities. They are derived from preimplantation or early post-implantation embryos, or reprogrammed from somatic cells. PSCs from various species, such as mice [1], humans [2], pigs [3], and cattle [4], have been established. Generally, PSCs can be categorized into primed and naïve PSC states, corresponding to the post-implantation and preimplantation epiblasts. In recent years, other new types of PSCs have been reported, such as expanded or extended potential stem cells (EPSCs) [3,4,5,6,7,8]. Generally, these PSCs are maintained in a medium cocktail modulating the developmentally important signaling pathways to minimize downstream differentiation but can generate various cell types when re-introduced into preimplantation embryos (in the case of mouse embryonic stem cells), or by directed or random differentiation in vitro. The PSCs have the ability to autonomously form structures resembling the blastocyst, called blastoid.

Human and pig pluripotent stem cells have emerged as promising tools in regenerative medicine. Compared to mice, pigs offer a more relevant animal model due to their genetic and physiological similarities to humans. Cellular products derived from human PSCs have shown potential in treating advanced-stage diabetes [9] and macular degeneration [10,11]. However, challenges such as immune compatibility, long-term functionality, and oncogenic transformation need to be carefully addressed for their clinical application.

In addition, the integration of human PSCs into pig embryos [12,13,14] and the use of pig PSCs for generating humanized pig organs through blastocyst complementation [15] or blastoid formation have opened up new avenues for organ transplantation. Although these approaches hold promise, they also pose numerous challenges that must be addressed to ensure their successful implementation. Overcoming issues such as apoptosis [16,17], developmental disparity [15,18], and pace variation [12,13,14] are crucial in order to overcome these hurdles.

Furthermore, pig PSCs provide a distinct and valuable platform for producing genetically modified animals specifically for xenotransplantation purposes. When compared to pig embryos or fibroblasts as cell sources for genetic engineering and the creation of humanized pigs, pig stem cells possess unique advantages. Pig stem cells exhibit higher efficiency in gene editing and offer the possibility of animal generation through various potential methods [3,19,20], such as germline transmission in chimera, cloning, and blastoid formation. Although these methods are still being validated, they hold significant potential in utilizing pig stem cells for xenotransplantation purposes.

This review aims to explore the application of human and pig PSCs in regenerative medicine, highlighting their potential in producing cellular products, tissues, and genetically modified animals, thereby underscoring their significant impact in the field.

## 2. Cellular Products

Human-induced pluripotent stem cells (iPSCs) and pluripotent stem cells (PSCs) can provide an almost limitless source of seeding cells for therapeutic purposes. For degenerative diseases, there is no curative therapy available. Therefore, researchers are exploring the possibility of replacing dysfunctional cells with stem cell-derived differentiated cells, with the aim of restoring physiological function and ensuring long-term safety in the human body. Currently, numerous trials are underway, testing human PSC-derived cellular products for clinical use. In this section, we will primarily discuss two regenerative medicine cases.

### 2.1. Advanced Stage Diabetes

For both type 1 and type 2 diabetes, at an advanced stage, pancreatic β cells are either completely destroyed or exhausted, leading to uncontrolled hyperglycemia. The transplantation or replacement of pancreatic β cells is a promising approach to restoring islet function and endogenous insulin secretion. Researchers have already successfully derived β cells with insulin secretion ability from human iPSC-derived definitive endoderm to pancreatic progenitors, by stepwise modulating signaling pathways, such as Activin A, SHH, FGF, NOTCH, and IGF [21]. Using these cells, in 2021, a VS-8803 trial was conducted in three T1DM patients, and all showed a good response with controlled blood glucose levels and good tolerance and safety [9]. Despite the safety and efficacy of VS-8803, the widespread clinical application of PSC-derived β cells still faces many challenges. For example, immune tolerance is not satisfactory, and without a long-term immunosuppressive regimen, human PSC-derived β cells will be attacked by the immune system. Encapsulating stem cell-derived β cells with a semi-permeable membrane allows hormone secretion while avoiding direct contact with immune cells [22]; alternatively, using PSCs expressing immune checkpoint CTLA4-Ig and PD-L1 or those universal and off-the-shelf PSCs may reduce immune cell attack [22]. The immune-tolerant β cells derived from iPSCs would facilitate their widespread use for the treatment of diabetes and other diseases.

### 2.2. Macular Degeneration

Another example of human PSC application is age-related degenerative disease. Macular degeneration (AMD), a leading cause of blindness in old people, is marked by dystrophy and dysfunction of the retinal pigment epithelium. A cell-based therapeutic approach is to replace the aged retinal pigment epithelium cells with the normal ones.

Recently, researchers succeeded in deriving retinal pigment epithelium from human PSCs. To achieve this, hPSCs were first differentiated to the neuroectodermal lineage by WNT, TGFβ and BMP signaling inhibition. After several days of neural induction, the differentiation medium was switched to an RPE induction recipe, activating the SMAD2, and GSK3β [23].

This human PSC-derived RPE has been tested in several clinical trials [10,11]. The current evidence from rodents [3,24], swine [10,11], rabbits [25], non-human primates [26], and human clinical trials [11,27,28] shows the medium-term safety and efficacy, which improve partial retina function and vision-related quality of life with no evidence of adverse events, such as immune rejections, tumor formations, or graft failure [11,27]. In the case of the allogeneic transplantation of HLA-mismatched hPSC-derived RPE, researchers also showed a 2-year survival of RPE with no evidence of immune reaction [11].

However, to achieve long-term treatment targets, challenges remain. Similar to all cell replacement therapies, there are concerns about immune rejection [29], long-term survival [30], aging and dysfunction, and the oncogenic transformation of transplanted cells [31,32,33,34].

On the other hand, the transplanted RPE may benefit from a privileged immune environment compared to cases like diabetes, due to the protection provided by the blood–retina barrier that does not recognize foreign antigens [29]. However, some studies have shown T-cell and NK-cell responses to human ESC-derived RPE [31], raising concerns about long-term immune compatibility. One approach to address this issue is the genetic removal of HLA classes I and II, allowing human ESC-RPE to escape T-cell attacks and reduce in vivo rejection [31].

Another challenge is to ensure the long-term functionality of PSC-derived RPE derived from stem cells. In order to reduce RPE apoptosis and oxidative stress during the peri-transplantation window, it is suggested to overexpress key factors related to normal function, such as pigment epithelium-derived factor (*PEDF*), anti-apoptotic genes like *BCL-2* and *XIAP*, or anti-inflammatory genes like *IL-10*.

Lastly, concerns have been raised about oncogenic transformations in all stem cell-based clinical products [31,32,33,34]. After long-term culture in vitro, cancer-related mutations can occur and accumulate at a rate 40-fold higher than in vivo cases [33,35]. To mitigate the risk of mutations, it is important to use early passage seed cells, regularly check cell morphology to identify transformed or unhealthy cells, and carefully examine genomic stability and molecular properties, RNA sequencing, or whole genome sequencing before transplantation [31]. Further studies are required to assess the biocompatibility of different methods for improving cellular product transplantation applications.

## 3. Human or Humanized Pig Organ Generation

In addition to cell products for regenerative medicine, recent studies have shown the potential to generate transplantable organs or tissues using human or large animal-derived PSCs (Figure 1).

Pigs, in particular, possess unique advantages for generating humanized organs and applications in regenerative medicine. Unlike non-human primates, they offer greater availability, lower maintenance costs, and fewer ethical concerns. Additionally, compared to bovine and sheep organs, pigs’ heart valves have long been used in clinical settings [36], thus exhibiting better immune tolerance and biosafety. Moreover, their genetic closeness as well as physiological and metabolic similarities to humans further support their utility in these applications.

Various approaches can be employed to generate human or humanized pig organs. One approach involves creating human–pig chimeras by integrating human PSCs into pre-implantation pig embryos [12,13,14], allowing for the further development and maturation of human cells in the pig recipient. This could potentially result in the harvesting of human tissues from chimeric piglets. Another approach involves using pig–pig chimeras [15], where humanized pig PSCs are integrated into pig embryos and undergo further development. Also, taking advantage of the full developmental potential of human and porcine PSC, it is also possible to produce a human organoid or humanized porcine organoid in a culture dish. Moreover, humanized pig PSCs could also serve as nuclear donors to generate fully humanized pigs via cloning [19,20]. In addition to cloning, pig blastoids generated from pig PSCs may offer a promising technology for obtaining genetically modified pigs for organ transplantation. However, these new approaches are still at various developing stages and need time to address technical challenges and regulatory hurdles before clinical applications.

### 3.1. Chimera Formation in Rodents PSC

The term “chimera” originally comes from Greek mythology, referring to a monster with a lion’s head, goat’s body, and snail’s tail. In biology, a chimera refers to a living organism with a mixed genetic background. In 1968, Richard Gardner et al. pioneered the mouse–mouse chimera assay by injecting mouse ESCs into a mouse blastocyst, resulting in mosaicism with pigmented eyes and coat color, demonstrating the contribution of mouse ESCs to various tissues [37]. Building on this principle, researchers began exploring interspecies chimeras with the goal of generating functional organs from another species, which was inspired by the concept of blastocyst complementation, where wild-type ESCs are injected into a host embryo with a specific gene mutation that causes developmental defects in a particular organ. The injected ESCs take advantage of the developmental niche provided by the host embryo and complement the agenesis defects. For example, when mESCs were injected into *pdx1* null rat embryos that lacked a pancreas, they efficiently formed a rat-sized pancreas with almost all cells derived from mouse donor cells [38]. These mESC-derived islets were then xenotransplanted into diabetic mice, effectively rescuing the disease and maintaining normal blood glucose levels without immune rejection. Since 2010, there have been numerous examples of mammalian organ generation using the blastocyst complementation principle, including mouse–mouse, rat–rat, mouse–rat, rat–mouse, or pig–pig chimeras. These experiments have shown clear evidence of generating functional and therapeutic organs, such as vascular/blood structures [39], kidney [40], heart [41], lung [42], functional thymus [43], pancreas [38,44], liver [45], retina [46], inner ears [47], germ cells [48], and more, which can be xenotransplanted into diseased animal models.

### 3.2. Chimera Formation in Animals Using Human and Pig PSC

The aforementioned successes open the feasibility of generating human organs via interspecies chimeras. However, the efficient integration of human PSCs into mouse, rat, rabbit, or large-animal (such as cattle or pig) blastocysts has proven to be challenging, still with limited success (Table 1).

#### 3.2.1. Apoptosis

One of the major obstacles is the tendency of human PSCs to undergo apoptosis when injected into early embryos [16,17]. To find the underlying mechanisms, Wu Jun and colleagues developed an in vitro co-culture model to study cell–cell competition in an interspecies context, using pluripotent stem cells from both humans and mice [17]. Through transcriptome analysis, the authors found that NF-κB signaling played a crucial role in inducing apoptosis in human cells when co-cultured with mouse cells.

Apart from apoptosis during cell competition, cell aging is another hurdle. Evidence showed that the poor quality of human PSCs in vitro was accompanied by progressively shortening telomeres, which was associated with cellular aging [12,16,17], self-renewal, and pluripotency [50,51]. Besides modifying key elements of NF-κB signaling, it might be helpful to either inactivate *P65*, *P53* or the transgenic expression of *MYD55*, *BMI1*, *BCL2* in human PSCs [12,13,16,17,52]. To avoid cellular aging, we can temporarily express *ZSCAN4* [53] or inhibit mTOR signaling [54]. Those strategies could potentially help enhance fitness and improve chimerism.

#### 3.2.2. Developmental Disparity

The second challenge was the developmental disparity between PSCs and host embryos. It is speculated that the microenvironment of the host embryo would eventually exclude the unmatched donor PSCs. For example, in intraspecies pig–pig chimeras, when porcine EpiESCs (derived from post-implantation embryos) were injected into porcine pre-implantation blastocysts, rarely any chimerism was observed after further in vivo development at E10 [15]. This is likely due to the fact that the donor pig EpiESCs and the host pig blastocyst were at different developmental time points, preventing proper synchronization between the cells and the host embryos. Similar phenomena have been observed in mouse EpiESCs, which showed low chimerism in mouse pre-implantation embryos, and similarly, mESCs also exhibited poor survival in post-implantation embryos [18]. Therefore, when introducing primed human PSCs, which resemble the post-implantation mouse epiblast stage [55], to a host embryo from a pre-implantation stage, low chimerism is expected due to this developmental disparity. On the other hand, naïve hPSCs have a more open chromatin structure and epiblast pigenetic signature, making them more developmentally plastic and easier to mature within the host embryo. Therefore, most human PSC chimeras have been tested using naïve or intermediate human cells, which exhibit higher efficiency [14]. However, large-scale analysis has revealed that naïve human PSCs are genome-wide hypomethylated [8,56], and their imprinting pattern is abnormal [57]. This loss of imprinting may prove to be carcinogenic and problematic for clinical applications.

Another type of human PSC, called human expanded potential stem cells (human EPSCs), may be an alternative developmental match for pre-implantation host embryos. These cells have been reported to have a higher DNA methylation level and are considered to be developmentally more primitive [3,4,6]. Given their genomic stability and molecular features, EPSCs may potentially serve as an alternative donor cell source for interspecies chimeras. In this regard, human EPSCs have been used for in vitro chimera experiments, including with mouse and monkey early-stage embryos [5,58]. Subsequent analysis of these chimeric embryos following in vitro culture has revealed embryonic and extraembryonic contributions from human EPSCs. However, further extensive testing and evaluation of the competency and chimerism of human EPSCs in extended culture or in vivo development are required.

#### 3.2.3. Developmental Pace Variations

The third difficulty lies in the variations in developmental pace. Although the expression of lineage marker genes and regulatory networks shows cross-species conservation in general, the timing of developmental events exhibits species-specific variation.

In mouse embryos, the first lineage segregation occurs early at E3.0. In contrast, in human and pig embryos, the separation of the inner cell mass (ICM) and trophectoderm (TE) occurs relatively late at around E5.0 to E6.0, followed immediately by the second wave of fate determination and the emergence of the epiblast and hypoblast at around E6.0. Consequently, mouse embryos progress much more rapidly than human and pig embryos, making them unsuitable as host embryos.

On the other hand, pigs are genetically close to humans and exhibit physiological and metabolic similarities, making them a potentially ideal host animal for generating human organs. The first published trial on human–pig chimeras was conducted in 2017 by Wu Jun et al., whereas human PSCs demonstrated chimeric contribution to monkey embryos until day 19 [14]. They also conducted two parallel experiments using rat naïve PSCs to test chimera contribution to two different host embryos: mouse embryos (developmentally close to rats) and pig embryos (developmentally distant from rats). The rat–mouse chimera showed good chimerism, while the rat–pig chimera showed no chimerism due to the phylogenetic distance. Currently, by using pigs as host embryos in combination with the blastocyst complementation principle, researchers have been able to generate humanized muscle and hematoendothelial cells from human PSCs [12,13].

## 4. Humanized Pig PSCs for Xenotransplantation

Apart from growing human PSCs in chimera to generate humanized organs, researchers are also trying to directly generate humanized pigs for organ transplantation. However, inter-species incompatibility poses a significant threat to future clinical xenotransplantation application. The long-term survival and normal function of xeno-grafts in the host have been greatly hindered by hyperacute rejection, acute vascular rejection, cellular rejection, and delayed xenograft rejection [59], which not only encompass the immune system but also elicit a very complex interaction of coagulation and inflammation response.

### 4.1. Brief History of Xenotransplantation

The history of xenotransplantation dates back to the 17th century with the successful blood transfusion from a lamb to a boy [60]. However, subsequent attempts using animal organs from bovine, goat, and pig proved largely ineffective and fatal. Despite the use of immunosuppressive drugs in the 1960s, chimpanzee-to-human heart transplants only survived for 90 min [61]. In the 1990s, researchers identified αGal as a major antigen mediating pig-to-human rejection [62]. Therefore, with advancements in gene editing and cloning techniques, the first genetically modified pig was generated in 2003 [20,63]. In the following years, more important genetic modification strategies were put forward, and genetically engineered pigs were generated. In 2022, successful pig-to-human heart [64] and kidney transplants [65,66] were reported and showcased the promising potential of using humanized pigs for organ transplantation.

### 4.2. Gene Modification Strategies

Until now, some important genetic loci have been discovered and edited in pig cells to mitigate the undesirable rejection effects. For example, the xeno-derived glycan epitopes (αGal, NeuG5c, Sd(a)-like glycan) would strongly activate the preformed antibodies in the host, causing organ failure within several minutes to hours. Scientists engineered the glycan epitopes to create the triple knock-out pig grafts [67], attempting to reduce transplant rejection. More recently, gene editing tools have enabled additional genetic manipulations, such as genes involved in complement activation (human CD46, human CD55, and human CD59), coagulation regulation (human thrombomodulin (TBM), human endothelial protein C receptor (EPCR), and human vWF), endothelial protection (heme oxygenase-1 (HO-1)), and immune modulation (macrophage inhibition-human CD47, NK cell inhibition-HLA-E, and HLA-G). In addition to using genetic engineering to address immune rejections, there are potential genetic methods to reduce the risk of zoonotic transmission from pigs to humans. One such approach involves the successful elimination of all porcine endogenous viruses (PERVs) in pigs through genetic manipulation, resulting in PERV-free pigs suitable for xenotransplantation [68]. Based on these, scientists have successfully created pigs with various combinations of targeted genetic changes using genome-editing techniques [69,70].

### 4.3. Cell Sources for Xenotransplantation

The cell source for generating multi-gene modified pigs could be pig fibroblasts, pig embryos, and pig embryonic or pluripotent stem cells (Figure 2). There are pros and cons for each cell source to generate genetically modified pigs (Table 2).

#### 4.3.1. Pig Embryos

The most direct approach involves gene editing in the pig zygotes. Exogenous gRNA and Cas9 protein can be introduced into pig zygotes through microinjection, electroporation, or lipofection [71], resulting in the generation of transgenic pigs with germline competency [58]. These offspring exhibit comparable health to their wildtype counterparts. However, there are certain limitations to this method. Firstly, it allows for only limited gene editing for xenotransplantation, thereby restricting the extent of genetic modifications that can be performed. Secondly, the analysis of gene editing outcomes can be challenging. The F0 founder pig may exhibit mosaicism across different tissues, leading to potential false positive or false negative results [72,73]. Additionally, crossbreeding F0 homozygous animals can sometimes result in undesired phenotypes in the F1 generation as observed in mouse studies [74]. Therefore, the direct gene editing of porcine zygotes can be time-consuming, less efficient and with unclear consequences.

#### 4.3.2. Pig Fibroblasts

Subsequently, as pig somatic cell nuclear transfer (SCNT) technology advanced, researchers turned to gene editing fibroblasts followed by cloning to generate humanized pigs [58]. Currently, using fibroblasts for gene editing is the most common method to obtain humanized pigs (Figure 2) [58]. Fibroblasts are easy to handle and also allow for gene editing. Researchers have even successfully generated 3KO-9Tg pigs (knockout of three pig xeno-genes and insertion of nine human genes) with germline transmission using porcine fibroblasts as the gene-editing source cells [70]. However, gene editing analysis of fibroblasts requires serial dilution to ensure clear genotyping results due to the presence of cell–cell mixtures. Furthermore, fibroblasts have limitations in terms of passage numbers, low genome-editing efficiency, and limited capacity for precision and complex gene targeting. Additionally, edited fibroblast editing does not permit immune compatibility testing and it requires SCNT to generate pig endothelial cells and organs for in vitro immune compatibility assessment or direct in vivo transplantation in non-human primate (NHP) models. This SCNT-based approach is time-consuming for selecting optimal combinations of pig gene edits and human gene insertions, which critically delays progress in the field.

#### 4.3.3. Pig Pluripotent Stem Cells

On the other hand, the emergence of pig embryonic or pluripotent stem cells (PSC) provided a promising donor cell source for xenotransplantation, addressing the above-mentioned limitations. Several types of pig PSCs are currently available, including pig EPSCs (expanded potential stem cells) [3], pig EpiSCs (epiblast-derived stem cells) [19], and pig EDSCs (embryonic disc stem cells) [20], derived from both pre-implantation and peri-implantation embryos. Notably, EPSCs are of preimplantation embryo origin, exhibit higher homologous recombination activity [75] and possess unlimited proliferation potential, enabling complex and precise gene editing. Additionally, EPSCs can differentiate into both embryonic and extra-embryonic tissues [3,15,20]. Furthermore, their ability to grow from single cells and form colonies facilitates unambiguous genotyping analysis and the generation of well-defined genetically modified stem cells.

These unique characteristics of pig pluripotent stem cells offer several potential applications. Firstly, they can serve as nuclear donors for generating cloned pigs. Secondly, they can potentially be utilized for generating humanized pig organs through blastocyst complementation and germline transmission in chimera assays. Lastly, pig stem cells can be used to create blastoids (artificial blastocyst counterparts) that can potentially be transferred into pig uteri to potentially generate pigs.

### 4.4. Generation of Humanized Pigs through Cloning Techniques

Pig PSCs are an ideal cell source in xenotransplantation. However, there is still little investigation related to their stem cell cloning efficiencies. Some very early studies showed that long-term cultured established mouse ESC lines could be used in generating cloned mice [70], which showed the cloning application of other species’ pluripotent stem cells, like the pig.

However, the road has not been easy for deriving cloned pigs from PSCs. Ten years ago, Lai et al. generated a total of 11,923 cloned embryos using pig iPSCs, but only resulted in 25 pregnancies at 24–26 days, and no cloned piglets were generated [76]. The reason behind this was probably due to the exogenous reprogramming factor gene expression in pig iPSCs, which hindered cloned embryo development.

More recently, pig PSCs derived from embryos have been generated, which could serve as nuclear donors for cloning at efficiencies similar to those using fibroblasts [15,20], where pig PSCs were pre-differentiated in a medium containing BMP4, SB431542, and FGF2 [15]. In the meantime, the epigenetic remodeling of the cloned embryo was found to improve the cleavage and blastocyst rates. Substances, such as scriptaid [77], abexinostat [78], vitamin C [79], trichostatin A (TSA) [80,81], cytochalasin B [20,81], Kdm4d/Kdm5b mRNA injection [81,82], and others could also be useful in PSC-based pig cloning. Furthermore, cell cycle synchronization by serum starvation or culture to confluency to arrest donor pig cells at the G0/G1 phase were found to further improve the cloning efficiency. Apart from this, the quality of the starting cells, particularly the donor cell chromosome stability, is an important factor. Checking the karyotyping and using early passage pig stem cells improved donor efficiency.

### 4.5. Acquisition of Humanized Pig Organs through Chimera Formation

As for using pig stem cells as donors for chimera assays to generate pig organs, many experiments are still under investigation. As stated above, scientists have already shown in mouse studies that mESC could give rise to fully functional mouse organs in organ agenesis mouse or rat embryos. It is foreseeable that through blastocyst complementation, genetically modified pig PSCs could also generate humanized pig organs in organ agenesis hosts. Until now, there have been no studies using pig multi-gene modified PSCs to generate humanized pig organs via blastocyst complementation in pig embryos. However, previous studies have paved the way for testing this idea. Scientists genetically knocked out *SALL1* in pig zygotes which revealed no kidney phenotype [83], knocked out *ETV2* which showed no endothelium [13], and removed *HHEX* which showed no liver phenotype [45]. Those pig genetically mutant agenesis embryos, combined with genetically edited pig PSCs, offer an attractive approach for the generation of humanized pig organs in transplantation.

### 4.6. Production of Humanized Pigs via Blastocyst-like Structure Development

Alternatively, the generation of humanized pigs via blastoid formation also holds great promise. Blastoids are stem cell-derived structures that resemble natural blastocysts, which can be generated by modulating specific signaling pathways in pluripotent stem cells. The first mouse blastoid was successfully created by Nicolas Rivron’s group in 2018 [84], by sequential aggregation of mESCs (mouse embryonic stem cells) and mTSCs (mouse trophoblast stem cells). These blastoids not only exhibited morphological similarities to E3.5 mouse blastocysts but also demonstrated functional capabilities by implanting into pseudo-pregnant mice and triggering endometrium decidualization.

Since then, rapid advancements have been made in blastoid generation [56,85,86,87,88]. It is now possible to directly reprogram somatic cells to blastoids [86] or even form a blastoid from a single type of pluripotent stem cell. Recent studies have achieved the generation of human blastoids with an efficiency of up to 70% through the triple inhibition of Hippo, Nodal, and Erk signaling pathways [87]. Although both mouse and human blastoids have limitations in their developmental potential, being unable to generate functional tissues or organs, these groundbreaking studies provide a promising outlook for regenerative medicine.

Looking ahead, the generation of pig blastoids from pig PSCs is plausible in the future. With further refinement of signaling pathways and culture conditions, it may be possible to develop multi-gene modified pig blastoids that can develop in an in vivo environment. While the authentic blastocyst model is still in its infancy, this innovative approach to generating living organisms holds tremendous potential for advancing developmental biology research and revolutionizing regenerative medicine.

## 5. Ethics and Other Concerns

There is a higher social acceptance of pig-related products for therapeutic purposes—they have a long history of domestication as livestock and serve as an important food source for human beings. This simple argument is, however, not a valid argument for xenotransplantation, as the research touches on many more ethical aspects. Therefore, xenotransplantation research is accompanied by ethical research to evaluate the perception of the different ethical groups regarding the use of pig organs. Pioneering work has been established by the German xenotransplantation consortium. In a survey of representative citizens, the group considered the benefits of xenotransplantation to outweigh the risks but called for strict regulatory measures [89]. Despite animals having been used in scientific research for a long time, the main ongoing concern is the welfare of these animals, ensuring that they live well, feel well, and function well [90]. Therefore, when manipulating the pig genome to create humanized pigs for xenotransplantation, it is essential to carefully assess gene modifications to avoid harming the animals’ physical and mental health.

When employing human PSCs for human-pig chimeras or directly utilizing humanized pig organs for regenerative medicine, we need to avoid human PSCs contributing to the neurons and germ cells. Efforts could be made to specifically eliminate the donor cell contribution in the brain in interspecies chimera [91]. Such scientific advances are expected to help circumvent the contribution of human cells to the pig brain or germ cells.

Again, the main argument for an overall positive assessment of stem cell research in general and xenotransplantation in particular in the societal context is primarily based on the societal obligation to help people with life-threatening conditions and to alleviate suffering where possible. The fact that research is well ahead of the public consciousness and that a broader societal debate is necessary and still lacking demands ongoing evaluation and regulation to ensure the responsible and safe implementation of using stem cell- and pig-related products in research and medicine. These research activities, including the generation of gene-edited pigs, must have ethical approval by an independent commission and abide by the local and international regulations.

## 6. Conclusions

The availability of PSCs from various species, particularly humans and pigs, has greatly expanded the potential applications of stem cells. Researchers have successfully derived a wide range of clinically relevant cell types from both human and pig PSCs. In addition, the generation of transplantable organs and tissues offers a potential solution to the global organ shortage. By utilizing techniques like chimera formation and blastocyst complementation, where stem cells are integrated into agenesis host embryos, it would be possible to generate human organs or humanized organs. Furthermore, pig PSCs are an ideal cell source for gene editing in xenotransplantation.

Overall, the advent in the generation of PSCs from multiple species has significantly expanded stem cell application possibilities, from generating specific cell types for therapeutic purposes to addressing organ shortages through innovative techniques like chimera formation and blastoids. However, these innovative approaches are still in their infancy and require further investigation.

## Figures and Tables

**Figure 1 cells-12-02075-f001:**
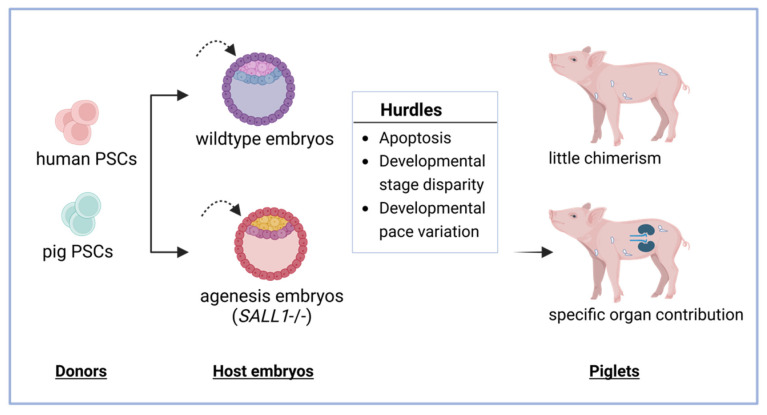
Organ generation from human or pig pluripotent stem cells. Human or pig pluripotent stem cells (PSCs) could generate functional organs via chimera formation. Combined with blastocyst complementation, specific organ contributions could be enriched. For example, *SALL1* is important for kidney development; the donor PSCs would compensate for the host’s developmental defects (no kidney), specifically enriched in the kidney. The same principles could be applied to other organs.

**Figure 2 cells-12-02075-f002:**
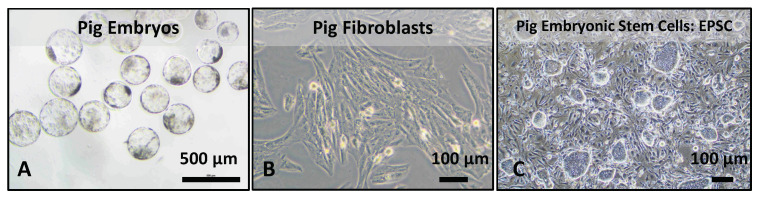
Cell sources for generating genetically engineered pigs in xenotransplantation. Brightfield images depict the potential cell sources for gene editing and the generation of humanized pigs for xenotransplantation. Pig embryos (**A**), fibroblasts (**B**), and embryonic stem cells (EPSCs) (**C**) are shown as viable options for obtaining genetically engineered pigs.

**Table 1 cells-12-02075-t001:** Summary of cross-species chimera using pluripotent stem cells.

Author	Donor–Host Relationship	Donor PSC Type	Donor Cell Culture Condition	Host Embryo Type	Lineage Contribution	PSC Contribution Percentage	Germline Transmission
Yang, Y. et al. (2017) [5]	Mouse–mouse	Mouse EPSCs	LCDM medium	Mouse 8C embryo, WT	At mouse E17.5, to both embryonic and extra-embryonic lineages	1% to 19%	Yes
	Human–mouse	Human EPSCs	LCDM medium	Mouse 8C embryo, WT	In vitro cultured mouse blastocyst, to both ICM or TE part	14.70%	No
Wu, J. et al. (2017) [14]	Rat–mouse	Rat naïve ESC	rat ESC medium	Mouse blastocysts	3-week-old rat-mouse chimera, to brain, heart, intestine, kidney, lung, pancreas, stomach, liver etc, no germline reported	From 0.1% to 10% across different tissues	Not reported
	Rodent–pig	Mouse/Rat naïve ESC	Mouse/rat ESC medium	Pig parthenogenesis blastocyst	At pig E21-28, no contribution	0%	No
	Human–pig	Naïve or intermediate human iPSC	Naïve culture: 2iLD-hiPSCs, NHSM-hiPSCs, 4i-hiPSCs, intermediate: FAC medium	Pig parthenogenesis blastocyst	At pig E21-28, all three germ layers. 25–45% embryos showed chimerism, among chimera embryos, 75% showed growth retardation.	Not reported	No
	Human–cattle			Cattle IVF blastocyst	In vitro cultured cattle embryo at E9, 60–80% embryos with chimerism to ICM, not to TE.	Not reported	No
Huang, K. et al (2018) [16]	Human–mouse	Human primed PSC, BMI1-overexpression	mTeSR1 medium	Mouse morula or blastocysts	At mouse E10.5, all three germ layers	7.50%	No
Zhi, M. et al. (2020) [15]	Human–mouse	Human naive PSC	mTOR inhibition pretreated; in 2iL plus insulin medium	Mouse blastocysts	At mouse E17.5, all three germ layers, red blood cells, liver, retina, not to germine	0.1% to 4%	No
Maeng, G. et al. (2021) [12]	Human–pig	Human-primed WT and TP53-/- iPSC	mTeSR1 or TeSR-E8 medium	Porcine MYF5-/-MYOD-/- MYF6-/- cloned morula	At pig E27, preferred contribution to the myogenic lineages (muscle), not to neural and germline	1:1000 to 1:100,000	No
Das, S. et al. (2021) [13]	Human–pig	Human WT or BCL2 overexpression iPSC	mTeSR1	Pig ETV2-null SCNT embryo	At pig E17-18, 81% embryos showed chimerism with BCL2-OE human cells; 52% with normal human cells. all endothelial cells with human origin	1:2000 in BCL2-OE hiPSC, Less than 1:10,000 in WT hiPSC	No
Tan, T. et al. (2021) [49]	Human–monkey	Human EPSCs	LCDM medium	Monkey in vitro cultured embryos	In vitro culture until D19, to epiblast and hypoblast, undetected to trophoblast lineage (highest at D9 with 2%).	D19, 6% epiblast; 4% to hypoblast, 5% to trophoblast.	No
Zhi, M. et al. (2022) [15]	Pig–pig	Pig EpiSC	3i/LAF medium	Porcine SCNT blastocysts at E5	At pig E10, no contribution	0%	No

**Table 2 cells-12-02075-t002:** Comparisons among different cell sources to generating genetically modified pigs.

	Embryos	Fibroblasts	Embryonic Stem Cells
Gene editing	One time editing	Simple editing	Complex editing
Genotyping	Mosaicism in nature, unclear results	Require serial dilution, clear results	Single colony genotyping, clear results
Passage number	No passage number	Limited, less than 20 passages	Unlimited passage number
Pig generation	In vivo production	SCNT	SCNT/ Blastoid, remain further validation
Off-spring healthy status	Healthy	Reprogramming issues	Unknown

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
