# Peer review of "Human and Pig Pluripotent Stem Cells: From Cellular Products to Organogenesis and Beyond"

_cells, 2023, doi:10.3390/cells12162075_

Round 1
Reviewer 1 Report
The review by Xuan et al. is a nice summary of some recent and historical work regarding PSCs. The work is relatively well written and provides some interesting information from key experiment. The figures are helpful and support the review well. Many readers would fine the review educational and focused on a specific topic. However, the work would benefit from addressing the following points:
1) What is the purpose/main point of the review? This should be mentioned in the abstract and touched upon throughout.
2) It seems like the title refers to a wider review in PSCs, however, the review is focused on humanized pig PSCs.
3) It would be beneficial to touch upon the ethical issues regarding chimeras and especially humanized PSCs and animals.
4) The titles of 4.3, 4.4, and 4.5 are a bit awkward mainly because of the use of the work "get" in the titles.
5) On lines 84-89 a sentence/content is repeated twice.
6) Word "summary" is not capitalized in section 5.
Most of the work is relatively well written, but a few awkward sentences do exist.
Reviewer 2 Report
In this review manuscript, the authors summarized the human-animal chimerism with PSCs derived from various animals, including pigs, monkeys, and rodents, for organogenesis and regenerative medicine. The manuscript is well-organized and a balanced update that covers general knowledge as well as the pros and cons of cell sources for interspecies chimera formation. The topic is timely and of interest to the field, but to be satisfactory for publication in Cells, the reviewer suggests the following relatively minor revisions:
1) Title: The term “Large Animal” should be replaced, as the manuscript predominantly focuses on PSCs derived from pigs.
2) Introduction: Towards the conclusion of the Introduction section, it would be beneficial to explicitly outline the scope of the manuscript.
3) Lines 84-89: Redundant. Please properly revise this paragraph.
4) Line 113: Prior to the second sentence, please provide a description of the advantages of using porcine cells for organ generation. Providing a brief history how porcine cells has been used for research on human medicine somewhere in the manuscript would also make the direction more refined.
5) Lines 24, 189, and 280: What does “EPSC” mean in each line indicated? Please carefully check and revise abbreviations in the manuscript.
6) Please incorporate some relevant information regarding cattle PSCs as a large animal cell source for interspecies chimerism due to shown in Table 2.
7) References must be numbered in order of appearance in the text.
8) There are no references #11, 20, 21, 35, 74, 75, 76, 77, 79, 80, 81, 83, 85, 86, 87, 88, and 89 in the text.
9) Please carefully check and correct the reference numbers in Table. 2.
